# Enhancement of High Harmonic Generation in Bulk Floquet Systems

Abhishek Kumar,[1] Yantao Li,[1] and Babak Seradjeh[1, 2, 3]

[1]*Department of Physics, Indiana University, Bloomington, Indiana 47405, USA*
[2]*IU Center for Spacetime symmetries, Indiana University, Bloomington, Indiana 47405, USA*
[3]*Quantum Science and Engineering center, Indiana University, Bloomington, Indiana 47405, USA*
(Dated: July 3, 2023)

We formulate a theory of bulk optical current for a periodically driven system, which accounts for the mixing of external drive and laser field frequencies and, therefore, the broadening of the harmonic spectrum compared to the undriven system. We express the current in terms of Floquet-Bloch bands and their non-adiabatic Berry connection and curvature. Using this expression, we relate spatio-temporal symmetries of the driven model to selection rules for current harmonics. We illustrate the application of this theory by studying high harmonic generation in the periodically driven Su-Schrieffer-Heeger model. At high frequencies and low field amplitudes, we find analytical expressions for current harmonics. We also calculate the current numerically beyond the high frequency limit and verify that when the drive breaks a temporal symmetry, harmonics forbidden in the undriven model become available. Moreover, we find significant enhancement in higher harmonics when the system is driven, even for low field amplitudes. Our work offers a unified Floquet approach to nonlinear optical properties of solids, which is useful for realistic calculations of high harmonic spectra of electronic systems subject to multiple periodic drives.

## I. INTRODUCTION

High harmonic generation (HHG) plays an important role in extreme ultraviolet and attosecond physics [1]. The HHG in gaseous system, first observed experimentally in 1961 [2], is usually understood within a three-step re-collision model [3]. More recently, with its experimental realization [4–11] and theoretical modeling of inter-band polarization and intra-band nonlinear current in solids [12–16], HHG has emerged as a useful nonlinear probe of electronic properties of condensed matter systems, including topological materials [17–26].

Along with this renewed interests in HHG, periodically driven quantum systems have gained attention in recent years, in part due to the high degree of control they can offer in experiments [27–59]. In contrast to their equilibrium counterparts, driven systems have rich nonequilibrium dynamics and can support nontrivial topology with no equilibrium analogue. Due to its inherently nonlinear character, it is important to study of the effect of external drive on HHG and its relationship with topology in periodically driven crystalline systems. To do so, we need a non-perturbative theoretical formulation of HHG in such systems.

In this paper, we aim to advance our understanding of the interplay among external periodic drive, topology, and HHG by employing the power of Floquet theory [60–62], applied to periodically driven solids in the bulk under intense light. We note that the external periodic field by itself may not generate a current: HHG requires the presence of an optical field. We derive an expression for optical current in a periodically driven quantum system in terms of the occupation of Floquet-Bloch bands and relate it to quantum geometrical quantities like the nonadiabatic Berry connection and curvature. Along the way, we reveal some subtleties in the relation between HHG spectra and nonequilirbium topology. We demonstrate

the utility of this formulation by obtaining HHG selection rules due to spatiotemporal symmetries. More importantly, our approach naturally extends recent work on Floquet linear response theory [63] to general frequency mixing between the drive and the probe fields, thus revealing a mechanism for enhancement of HHG spectra in periodically driven systems.

As a concrete illustration, we study HHG spectra in the one-dimensional driven Su-Shrieffer-Higger (SSH) lattice model [43], which admits various trivial and Floquet topological phases. We confirm the general HHG selection rules in this model and demonstrate that by breaking the temporal part of a spatio-temporal symmetry, one can obtain high harmonics that are forbidden in the undriven model. This is only possible in the driven system. We also obtain analytical expressions for the optical current and HHG spectra in the high frequency approximation. Assuming ideal and projected occupations of Floquet-Bloch bands, numerical calculation shows that there is a significant enhancement in higher harmonics when the system is driven, even at lower field amplitudes. This means HHG can be generated even at low pulse intensities in a Floquet system.

The paper is organized as follows. In Sec. II, along with a primer on Floquet theory, we present a general formulation of optical current in a periodically driven system and discuss its selection rules. In Sec. III, we consider the periodically driven SSH model and apply our formalism to obtain the optical current and its high harmonics. In Sec. IV, we present and discuss numerical calculations of HHG spectra in the driven SSH model. We conclude in Sec. V with a summary and outlook. Some details of our calculations are provided in two Appendices. Throughout the paper, we shall use the natural units $\hbar = e/c = a = 1$, where $a$ is the relevant length scale, such as the lattice spacing.

## II. OPTICAL CURRENT IN FLOQUET SYSTEMS

### A. Primer on Floquet theory

For a time-periodic Hamiltonian, $\hat{H}(t) = \hat{H}(t + T)$, with period $T$, the solution of Schrödinger equation takes the form $|\Psi_\alpha(t)\rangle = e^{-i\Xi_\alpha t} |\Phi_\alpha(t)\rangle$, where $\Xi_\alpha \in (-\pi/T, \pi/T)$ is the quasienergy, a conserved quantity and the Floquet state, $|\Phi_\alpha(t)\rangle$, is also periodic in time with the same period $T$, and satisfies the Floquet-Schrödinger equation

$$[\hat{H}(t) - i\partial_t] |\Phi_\alpha(t)\rangle = \Xi_\alpha |\Phi_\alpha(t)\rangle. \quad (1)$$

Also, the time-ordered evolution operator

$$\hat{U}(t, t_0) := \text{Texp}\left[-i \int_{t_0}^t \hat{H}(s)ds\right] \quad (2)$$

$$= e^{-i(t-t_0)\hat{H}_F(t)} \hat{P}(t, t_0), \quad (3)$$

is decomposed into a periodic micromotion operator

$$\hat{P}(t, t_0) = \hat{P}(t + T, t_0) = \hat{P}(t, t_0 + T)$$

$$\equiv \sum_\alpha |\Phi_\alpha(t)\rangle \langle \Phi_\alpha(t_0)| \quad (4)$$

and the evolution under the Floquet Hamiltonian

$$\hat{H}_F(t) = \sum_\alpha \Xi_\alpha |\Phi_\alpha(t)\rangle \langle \Phi_\alpha(t)|. \quad (5)$$

For a Hamiltonian that depends on two commensurate frequencies $(\omega, \Omega)$, we can define $\gamma = \omega t$ and $\Gamma = \Omega t$ so that $\hat{H}(\gamma, \Gamma) = \hat{H}(\gamma + 2\pi, \Gamma) = \hat{H}(\gamma, \Gamma + 2\pi)$. We can then write the multi-mode Floquet-Schrödinger equation

$$\left[\hat{H}(\gamma, \Gamma) - i(\omega\partial_\gamma + \Omega\partial_\Gamma)\right] |\Phi_\alpha(\gamma, \Gamma)\rangle = \Xi_\alpha |\Phi_\alpha(\gamma, \Gamma)\rangle, \quad (6)$$

where $|\Phi_\alpha(\gamma, \Gamma)\rangle$ is periodic in $\gamma$ and $\Gamma$ with period $2\pi$.

We may expand any such function, $f(\gamma, \Gamma)$ in Fourier modes, $f(\gamma, \Gamma) = \sum_{m,M} e^{-im\gamma - iM\Gamma} f^{(m,M)}$ with

$$f^{(m,M)} = \oint e^{im\gamma + iM\Gamma} f(\gamma, \Gamma) \frac{d\gamma}{2\pi} \frac{d\Gamma}{2\pi}, \quad (7)$$

where $\oint$ is over a cycle of each integration variable. The single-time Schrödinger equation and its solutions are obtained simply by replacing $\gamma = \omega t$ and $\Gamma = \Omega t$ in the multi-mode Schrödinger equation and its solutions.

### B. Optical current and its harmonics

We now derive an expression for optical current in a time periodic system using Floquet theory. The density matrix of the system at some initial time $t_0$ is $\hat{\rho}(t_0)$, which evolves in time through a unitary operator $\hat{U}(t, t_0)$ as $\hat{\rho}(t) = \hat{U}(t, t_0)\hat{\rho}(t_0)\hat{U}^\dagger(t, t_0)$.

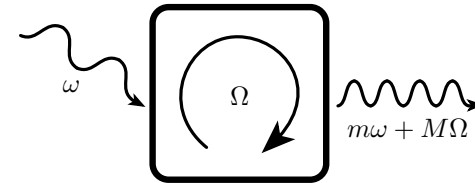

FIG. 1. Sketch for high harmonic generation in a Floquet system. The system is driven by a drive frequency $\Omega$ and irradiated by field frequency $\omega$. Here the output frequency is $m\omega + M\Omega$.

For further simplification, we assume that all the operators can be written in the single-particle fermionic basis and take periodic boundary conditions. So, the Hamiltonian $\hat{H}(t) = \sum_{krs} \hat{c}_{kr}^\dagger h_{rs}(k, t)\hat{c}_{ks}$, where $\hat{c}_{kr}^\dagger$ and $\hat{c}_{kr}$ are the creation and annihilation operators, respectively, at lattice momentum $k$ and quantum number $r$, and $h(k, t)$ is the time-periodic Bloch Hamiltonian. Similarly, the current operator $\hat{J}(t) = \sum_{krs} \hat{c}_{kr}^\dagger j_{rs}(k, t)\hat{c}_{ks}$, where $j(k, t) = \partial h(k, t)/\partial k$.

The current is

$$J(t, t_0) = \text{Tr}\left[\hat{\rho}(t)\hat{J}(t)\right] \quad (8)$$

$$= \sum_k \text{tr}\left[g(k)j^H(k, t, t_0)\right], \quad (9)$$

where $g_{rs}(k, t_0) = \text{Tr}[\hat{\rho}(t_0)c_{ks}^\dagger c_{kr}]$, 'Tr' is taken over the many-body Hilbert space, and 'tr' is over the single-particle Hilbert space for each momentum $k$. Here, $j^H(k, t, t_0) = u^\dagger(k, t, t_0)j(k, t)u(k, t, t_0)$ is the (single-particle) current in the Heisenberg picture and $u(k, t, t_0) = \text{Texp}[-i \int_{t_0}^t h(k, s)ds]$. In the Floquet basis $|\phi_\alpha(k, t)\rangle$ of $h(k, t)$, we find

$$J(t, t_0) = \sum_{k\alpha\beta} g_{\alpha\beta}(k, t_0)j_{\beta\alpha}^F(k, t)e^{-i(\epsilon_\alpha - \epsilon_\beta)(t-t_0)}, \quad (10)$$

where $j_{\beta\alpha}^F(k, t) = \langle \phi_\beta(k, t)| j(k, t) |\phi_\alpha(k, t)\rangle$ is periodic in time, and $\epsilon_\alpha$ are the single-particle quasienergies.

In the following, we specify the time dependence of the Hamiltonian through a periodically driven internal parameter $\delta(t) = \sum_M \delta^{(M)}e^{-iM\Omega t}$ and an external electric field via the gauge potential $A(t) = \sum_m A^{(m)}e^{-im\omega t}$. Thus, the Hamiltonian $h(k, t)$ depends on the two frequencies $\Omega$ of the drive and $\omega$ of the external field. Therefore, we may employ the Fourier expansion $j_{\beta\alpha}^F(k, t) = \sum_{m,M} j_{\beta\alpha}^{(m,M)}(k)e^{-i(m\omega + M\Omega)t}$ to write

$$J(t, t_0) = \sum_{\substack{k\alpha\beta \\ mM}} g_{\alpha\beta}(k, t_0)j_{\beta\alpha}^{(m,M)}(k)e^{-i(m\omega + M\Omega)t}$$

$$\times e^{-i(\epsilon_\alpha - \epsilon_\beta)(t-t_0)}. \quad (11)$$

The dependence on the initial time $t_0$ in Eqs. (10) and (11), where we set the occupation of the quasi-energy

bands, is the result of our assumption that the ensuing dynamics is fully coherent. However, in reality the occupation of the quasienergy bands, $g_{\alpha\beta}(k, t_0)$, depends on a specific relaxation process for a given system. We are interested in situations in which the relaxation process, e.g. through suitable coupling to a thermal bath, results in a diagonal population $g_{\alpha\beta} = g_\alpha \delta_{\alpha\beta}$ independent of $t_0$. This removes the dependence of the optical current on the initial time. Physically, this means the system approaches a Floquet steady state. Formally, we can obtain this result by assuming $g_{\alpha\beta}$ is independent of $t_0$ and average the current over $t_0$, to which only the diagonal occupations in Floquet states contribute.

Then the current becomes periodic in time $J(t) = J(t + T)$, and

$$J(t) = \sum_{k\alpha m M} g_\alpha(k) j_{\alpha\alpha}^{(m,M)}(k) e^{-i(m\omega + M\Omega)t}. \qquad (12)$$

This form of the optical current clearly shows the frequency mixing in the generation of its harmonics. For example, if the drive frequency is an integer multiple of the field frequency, $\Omega = N\omega$, the current acquires harmonics $m + NM$ of the field frequency. Conversely, when $\omega = n\Omega$ for integer $n$, the current acquires fractional harmonics $m + M/n$ of the field frequency. As we shall demonstrate in concrete examples, some of these harmonics may vanish or be enhanced depending on the order of the process and selection rules imposed by spatiotemporal symmetries.

We also note that this diagonal optical current may appear to correspond to only the intra-band contributions. However, it is important to bear in mind that each quasienergy band is obtained by a reconstruction of the original energy bands of static system. Therefore, the optical current, Eq. (12), already includes both intra- and inter-band contributions of the static system.

## C. Optical current from Berry connection and curvature

We would like to recast the optical current in terms of geometrical quantities like Berry curvature and Berry connection. First, we note that in the Floquet basis, the current matrix elements,

$$\begin{aligned} j_{\beta\alpha}^F(k,t) &\equiv \langle \phi_\beta(k,t)| \frac{\partial h(k,t)}{\partial k} |\phi_\alpha(k,t)\rangle \\ &= [\epsilon_\beta(k) - \epsilon_\alpha(k) + i\partial_t] \langle \phi_\beta(t)|\partial_k \phi_\alpha(k,t)\rangle \\ &\quad + \delta_{\alpha\beta}\partial_k \epsilon_\beta(k). \end{aligned} \qquad (13)$$

So the full expression of the current, Eq. (10), with diagonal Floquet population becomes

$$J(t) = \sum_{k\alpha} g_\alpha(k) \left[ \partial_t \mathscr{A}_{\alpha,k}(k,t) + \partial_k \epsilon_\alpha(k) \right], \qquad (14)$$

where $\mathscr{A}_{\alpha,k}(k,t) = \langle \phi_\alpha(k,t)|i\partial_k \phi_\alpha(k,t)\rangle$ is the Berry connection of the Floquet band $\alpha$. This expression makes it

clear that all time dependence and, therefore, all higher harmonics of the optical current result from the time dependence of the Floquet Berry connection. As in the static case, this also shows that a completely filled Floquet band cannot contribute to DC current.

On the other hand, the gauge invariance of the optical current is not manifested in Eq. (14), since the Berry connection $\mathscr{A}_{\alpha,k}(k,t)$ itself is not a gauge-invariant quantity. Therefore, we now further recast this expression in terms of the gauge-invariant Floquet Berry curvature,

$$\begin{aligned} \mathscr{E}_\alpha &= \partial_t \mathscr{A}_{\alpha,k} - \partial_k \mathscr{A}_{\alpha,t}, \qquad &(15) \\ &= i \langle \partial_t \phi_\alpha | \partial_k \phi_\alpha \rangle - i \langle \partial_k \phi_\alpha | \partial_t \phi_\alpha \rangle \qquad &(16) \end{aligned}$$

where $\mathscr{A}_{\alpha,t}(k,t) = \langle \phi_\alpha(k,t)|i\partial_t \phi_\alpha(k,t)\rangle$ is the non-adiabatic (Aharonov-Anandan) connection of the states in Floquet band $\alpha$ [64]. Numerically, we can also calculate the Berry curvature more efficiently and accurately without the need of fixing a special gauge [65].

To proceed, note that

$$\begin{aligned} \partial_t \mathscr{A}_{\alpha,k} &= \mathscr{E}_\alpha + \partial_k \langle \phi_\alpha | i\partial_t \phi_\alpha \rangle \qquad &(17) \\ &= \mathscr{E}_\alpha + \partial_k \langle \phi_\alpha | h | \phi_\alpha \rangle - \partial_k \epsilon_\alpha. \qquad &(18) \end{aligned}$$

Therefore, we find

$$J(t) = \sum_{k\alpha} g_\alpha(k) \left[ \mathscr{E}_\alpha(k,t) + \partial_k \eta_\alpha(k,t) \right], \qquad (19)$$

where $\eta_\alpha(k,t) = \langle \phi_\alpha(k,t)| h(k,t) |\phi_\alpha(k,t)\rangle$ is the expectation value of the instantaneous Hamiltonian in the Floquet band $\alpha$. As before, due to periodicity in crystal momentum $k$, this last term does not contribute in a completely filled band. However, in partially filled bands, it can contribute to DC as well as the higher harmonics of optical current.

We note that these expressions can be readily extended to higher dimensions,

$$\begin{aligned} \mathbf{J}(t) &= \sum_{\mathbf{k}\alpha} g_\alpha(\mathbf{k}) \left[ \partial_t \mathscr{A}_{\alpha,\mathbf{k}}(\mathbf{k},t) + \partial_{\mathbf{k}} \epsilon(\mathbf{k}) \right] \qquad &(20) \\ &= \sum_{\mathbf{k}\alpha} g_\alpha(\mathbf{k}) \left[ \boldsymbol{\mathscr{E}}_\alpha(\mathbf{k},t) + \partial_{\mathbf{k}} \eta(\mathbf{k},t) \right], \qquad &(21) \end{aligned}$$

where $\boldsymbol{\mathscr{E}}_\alpha = \partial_t \mathscr{A}_{\alpha,\mathbf{k}} - \partial_{\mathbf{k}} \mathscr{A}_{\alpha t}$ and the Floquet Berry connection $\mathscr{A}_{\alpha,\mathbf{k}} = \langle \phi_\alpha | i\partial_{\mathbf{k}} \phi_\alpha \rangle$ and $\mathscr{A}_{\alpha,t} = \langle \phi_\alpha | i\partial_t \phi_\alpha \rangle$.

## D. Selection rules

Here, we analyze the constraints on the current and its harmonics due to symmetries for the ideal Floquet occupation with each band either fully occupied or empty. There is a significant body of literature on selection rules due to symmetries [66–69]. In this work, we are interested in discrete spatio-temporal symmetries. In particular, we consider a unitary mirror ($I_F$) and an antiunitary time-reflection ($\Theta_F$) symmetries, which include time

glide operations,

$$h(k,t) = I_F^\dagger h(-k, t+t_I) I_F, \qquad (22)$$

$$h(k,t) = \Theta_F^\dagger h(-k, -t+t_R) \Theta_F. \qquad (23)$$

Under $I_F$ symmetry, the occupied Floquet bands $\alpha \in$ occ satisfy $I_F |\phi_\alpha(k,t)\rangle = \sum_{\beta \in \text{occ}} \mathfrak{I}_{\beta\alpha}(k,t) |\phi_\beta(-k, t+t_I)\rangle$, where $\mathfrak{I}(k,t)$ is a unitary matrix. In Appendix A, we show that this leads to a relation between the current at different times,

$$J(t) = -J(t+t_I). \qquad (24)$$

Therefore, the current vanishes at an odd number of times in any interval of length $t_I$ in the cycle, at the boundary of which the current is nonzero. For $t_I = \pi/\omega$, we get the following condition for the harmonics of the current,

$$J^{(p)} = (-1)^{p+1} J^{(p)}, \qquad (25)$$

which means in the presence of $I_F$ symmetry, only odd harmonics will survive.

Under $\Theta_F$ symmetry, the occupied Floquet bands satisfy $\Theta_F |\phi_\alpha(k,t)\rangle = \sum_{\beta \in \text{occ}} \mathfrak{T}_{\beta\alpha}(k,t) |\phi_\alpha(-k, -t+t_R)\rangle$, where $\mathfrak{T}(k,t)$ is an orthogonal matrix. As shown in Appendix A, this also leads to a relation between current at different times,

$$J(t) = -J(-t+t_R). \qquad (26)$$

As a result, the current must vanish at $J(t_R/2) = J(T/2 + t_R/2) = 0$. For $t_R = \pi/\omega$ we have the following properties of the harmonics of the current,

$$J^{(p)} = (-1)^{p+1} J^{(-p)}, \qquad (27)$$

which means the odd harmonics of the current will be real and even harmonics of the current will be imaginary, and the DC component $J^{(0)} = 0$. If $t_R = 0$ instead, the current $J(t) = -J(-t)$ is an odd function of time and we find

$$J^{(p)} = -J^{(-p)}, \qquad (28)$$

which means $J^{(0)} = 0$ and all the other harmonics will be imaginary.

When both $I_F$ and $\Theta_F$ symmetries are present, and taking $t_I = t_R = \pi/\omega$, we have

$$J^{(p)} = J^{(-p)} = (-1)^{p+1} J^{(p)}, \qquad (29)$$

which means that only odd harmonics are nonzero and they are real.

### E. Relation to topology

The expressions of the current in terms of the Floquet Berry connection and curvature, Eqs. (14) and (19), provide a link between the optical current and the quantum

geometry of Floquet bands. This geometry is also responsible for nontrivial topological properties of the system [70]. Here, we examine the connection between the optical current and nontrivial topology in different frequency regimes.

In the low-frequency limit, the Floquet states $|\phi_\alpha(k,t)\rangle$ approach, up to a smooth gauge [71], the adiabatic states $|\psi_\alpha^{\text{ad}}(k,t)\rangle$, i.e. the instantaneous eigenstates of the driven Hamiltonian, $h(k,t) |\psi_\alpha^{\text{ad}}(k,t)\rangle = E_\alpha(k,t) |\psi_\alpha^{\text{ad}}(t)\rangle$, assuming the instantaneous energy bands $E_\alpha(k,t)$ remain gapped. Then, the optical current of fully occupied bands approaches $J(t) \to \sum_{\alpha \in \text{occ}} \oint \mathscr{E}_\alpha^{\text{ad}}(k,t) dk$, where $\mathscr{E}_\alpha^{\text{ad}}(k,t)$ is the adiabatic Berry curvature. This is, of course, the Thouless pump, in which a charge

$$Q = \oint J(t) dt = \sum_{\alpha \in \text{occ}} \oint \mathscr{E}_\alpha^{\text{ad}}(k,t) dk dt \equiv \text{Ch}_{\text{occ}}^{\text{ad}}, \quad (30)$$

equal to the Chern number $\text{Ch}_{\text{occ}}^{\text{ad}}$ of the occupied adiabatic bands in $(k,t)$ space, is pumped in each cycle.

While Eqs. (14) and (19) hold formally at all frequencies, the gap structure of Floquet bands deviates significantly away from the adiabatic limit. As a result, the Floquet bands in the intermediate frequency range become partially filled. In the high frequency limit, other approximations such as rotating-wave frame and Floquet Magnus expansion become available. In certain cases, one can also justify or specifically engineer the Floquet bands to be be fully occupied or empty. For example, for high frequencies that are off-resonant with gapped static bands, one may take the occupation of Floquet bands to be nearly the same as the original static bands. In such case, the optical current is obtained from the non-adiabatic Berry connection and curvature of Floquet bands.

We would like to note a subtlety related to the choice of gauge in using Eqs. (14) and (19). Consider a driven one-dimensional Hamiltonian $h(k,t)$ with time-reflection symmetry, e.g. $h(k,-t) = h(k,t)$, and chiral symmetry $\{C, h(k,t)\} = 0$ satisfying $C = C^\dagger = C^{-1}$. Then, the Floquet Hamiltonians $h_F(k,t_*)$ at time-reflection symmetric times $t_* = 0, T/2$, inherit the chiral symmetry $C$ [72]. The topology of the Floquet bands is then characterized by two gauge-invariant winding numbers $w(t_*)$,

$$w(t_*) = \frac{1}{\pi} \sum_{\alpha \in \text{occ}} \oint \langle C\phi_\alpha(k,t_*)|i\partial_k \phi_\alpha(k,t_*)\rangle dk \in \mathbb{Z}. \tag{31}$$

There is a bulk-boundary correspondence [72] between $w(t_*)$ and the number of midgap bound states, $\nu_0$ and $\nu_\pi$, respectively, at quasienergies 0 and $\pi/T$,

$$\nu_0 = \frac{w(T/2) + w(0)}{2}, \quad \nu_\pi = \frac{w(T/2) - w(0)}{2}. \tag{32}$$

It is possible to choose a smooth gauge at each $t_*$, $|\phi_\alpha(k,t_*)\rangle \to e^{i\Lambda_\alpha(k,t_*)} |\phi_\alpha(k,t_*)\rangle$, such that

$$w(t_*) = \frac{1}{\pi} \sum_{\alpha \in \text{occ}} \oint \langle \phi_\alpha(k,t_*)|i\partial_k \phi_\alpha(k,t_*)\rangle dk.$$

It is then tempting to related $w(T/2) - w(0)$ to $\int_0^{T/2} J(t)dt$. However, while both of these quantities are gauge-invariant, this would not be in general correct because it is not in general possible to find a gauge $|\phi_\alpha(k,t)\rangle \rightarrow e^{i\Lambda_\alpha(k,t)}|\phi_\alpha(k,t)\rangle$ that smoothly connects $\Lambda_\alpha(k,0)$ to $\Lambda_\alpha(k,T/2)$ in the cycle. The degree to which this smooth gauge condition is broken is indeed what $\nu_\pi$ quantifies.

## III. HHG IN PERIODICALLY DRIVEN SU-SCHRIEFFER-HEEGER MODEL

### A. Model and symmetries

In order to illustrate the physics of high harmonic generation in Floquet systems concretely, we consider the Su-Schrieffer-Heeger (SSH) model with time-periodic hopping amplitudes. In addition, an external time-periodic and spatially uniform gauge field, $A(t)$, is introduced via Peierls substitution $k \rightarrow k - A(t)$. The full Bloch Hamiltonian is

$$h(k,t) = \mathsf{w}\left\{1 + m(t) + [1 - m(t)]\cos[k - A(t)]\right\}\sigma_x$$
$$+ \mathsf{w}[1 - m(t)]\sin[k - A(t)]\sigma_y. \quad (33)$$

Here, $\mathsf{w}$ is the average hopping amplitude. In the following, we take the driven field $A(t) = A_0 \sin(\omega t + \theta)$ and the driven hopping modulation $m(t) = m_0 + m_1 \cos(\Omega t)$.

The static SSH model ($m_1 = 0, A_0 = 0$) has a trivial ($m_0 > 0$) and a topological phase ($m_0 < 0$) phase protected by the chiral symmetry, $\{h, C\} = 0$ with $C = \sigma_z$. In addition, the static model has both unitary inversion (or mirror) symmetry ($I = \sigma_x, k \rightarrow -k$) and antiunitary time-reversal symmetry ($\Theta = K, k \rightarrow -k$, where $K$ is the complex conjugation).

In the driven model, these symmetries are in general lifted by the external field. However, for certain drive protocols, we may recover these symmetries upon a suitable mapping within the cycle. That is, the driven system may have related *spatiotemporal* symmetries, in which the temporal part of the Hamiltonian is also transformed. Indeed, the inversion symmetry can be restored for the driven system if there exists a time glide $\tau(t_I): t \mapsto t + t_I$ within the cycle, for which $A(t + t_I) = -A(t)$ is odd and $m(t + t_I) = m(t)$ is even. Similarly, the time-reversal symmetry is restored if there exists a reflection time $t_R$, for which $A(t_R - t) = -A(t)$ is odd and $m(t_R - t) = m(t)$

is even. The spatiotemporal symmetries of the driven system are, then,

$$I_F^{-1}h(-k, t+t_I)I_F = h(k,t), \quad I_F = \tau(t_I)I, \quad (34a)$$

and

$$\Theta_F^{-1}h(-k, -t+t_R)\Theta_F = h(k,t), \quad \Theta_F = \tau(t_R)\Theta. \quad (34b)$$

When the drive frequency is an even multiple of the field frequency ($\Omega = 2n\omega$, $n \in \mathbb{Z}$), we may choose $t_I = \pi/\omega$ for any $\theta$ to obtain symmetry under $I_F$, yielding $J(t+T/2) = -J(t)$. For $\theta = \pm\frac{\pi}{2}$, we can take $t_R = \pi/\omega$ to obtain symmetry under $\Theta_F$, whereby $J(t + T/2) = -J(-t)$. Similarly, if $\theta = 0$ or $\pi$, we can take $t_R = 0$ to find $J(t) = -J(-t)$. Therefore, selection rules for the ideal occupation of the Floquet bands apply to the harmonics of the current in these cases as discussed in Sec. II D.

### B. Current in the high frequency limit

In this section we derive an analytical expression for the current in diagonal Floquet occupation using high frequency approximation. We will calculate the Floquet Hamiltonian and states as well as micromotion operators using high frequency expansion. We assume $\Omega = N\omega$ for an integer $N$.

In the high frequency limit, the Floquet Hamiltonian and micromotion operator are

$$h_F = h^{(0)} + \sum_{n>0}\frac{[h^{(-n)}, h^{(n)}]}{n\omega} + O\left(\frac{1}{\omega^2}\right), \quad (35)$$

$$P(t) = \exp\left[i\sum_{n>0}\frac{h^{(-n)}e^{in\omega t} - h^{(n)}e^{-in\omega t}}{in\omega} + O\left(\frac{1}{\omega^2}\right)\right], \quad (36)$$

respectively, where $h^{(n)}(k)$ are the Fourier components of the Bloch Hamiltonian $h(k,t)$,

$$h^{(0)}(k) = [(1 + m_0) + \text{Re}[f_0(k)]]\sigma_x - \text{Im}[f_0(k)]\sigma_y, \quad (37)$$

$$h^{(n)}(k) = \frac{f_{-n}(k) + f_n^*(k)}{2}\sigma_x + i\frac{f_{-n}(k) - f_n^*(k)}{2}\sigma_y, \quad (38)$$

where,

$$f_n(k) = \frac{m_1}{2}(\delta_{n,N} + \delta_{n,-N}) + e^{-ik}\left\{(1-m_0)\mathscr{J}_n(A_0)e^{in\theta} - \frac{m_1}{2}\left[\mathscr{J}_{n-N}(A_0)e^{i(n-N)\theta} + \mathscr{J}_{n+N}(A_0)e^{i(n+N)\theta}\right]\right\}, \quad (39)$$

and $\mathscr{J}_n$ are Bessel functions. Thus, we may write the Floquet Hamiltonian as $h_F(k) = \mathbf{d}_F(k) \cdot \boldsymbol{\sigma}$, with

$$\mathbf{d}_F(k) = \left((1 + m_0) + \text{Re}[f_0(k)], -\text{Im}[f_0(k)], \sum_{n>0}\frac{|f_n(k)|^2 - |f_{-n}(k)|^2}{n\omega}\right), \quad (40)$$

and the micromotion as $P(k,t) = \exp[i\mathbf{V}(k,t)\cdot\boldsymbol{\sigma}]$, with

$$\mathbf{V}(k,t) = \sum_{n>0}\frac{1}{n\omega}\left(\text{Im}[f_n^+(k,t)], \text{Re}[f_n^-(k,t)], 0\right), \quad (41)$$

where $f_n^\pm(k,t) = [f_{-n}^*(k) \pm f_n(k)]e^{in\omega t}$.

| $N = \Omega/\omega$ | harmonic $p$ | $\tilde{W}_1$ | $\tilde{W}_2$ | $\tilde{W}_3$ | $\tilde{W}_4$ |
|---|---|---|---|---|---|
| even | odd | $iW_1$ | $iW_3$ | $iW_2$ | $iW_3$ |
| even | even | $W_2$ | $W_4$ | $W_2$ | $W_4$ |
| odd | odd | $iW_1$ | $iW_3$ | $W_2$ | $W_4$ |
| odd | even | $W_2$ | $W_4$ | $iW_1$ | $iW_3$ |

TABLE I. Choice of $\tilde{W}_j$ in Eq. (53), in terms of $W_j$, Eqs. (49).

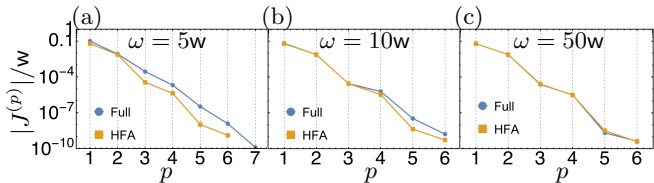

FIG. 2. Comparison of current harmonics obtained using the analytical expression in the high frequency approximation ("HFA") in Eq. (53) with the full numerical calculation ("Full"). The parameters are chosen as $m_0 = 0.3, m_1 = 0.17, A_0 = 0.1, \Omega = \omega$, and $\theta = 0$.

We can now use Eq. (10) to calculate the optical current by writing

$$j_{\alpha\alpha}^F = \langle \phi_\alpha(k,0)|P^\dagger(k,t)\frac{\partial h(k,t)}{\partial k}P(k,t)|\phi_\alpha(k,0)\rangle, \quad (42)$$

where $|\phi_\alpha(k,0)\rangle$ are the eigenstates of the Floquet Hamiltonian $h_F(k)$. The current operator $\partial h/\partial k = \mathbf{j}\cdot\boldsymbol{\sigma}$, where

$$\mathbf{j}(k,t) = w[1-m(t)]\Big(\sin[A(t)-k], \cos[k-A(t)], 0\Big). \quad (43)$$

Then, $P^\dagger(k,t)[\partial h(k,t)/\partial k]P(k,t) = \mathbf{j}_F \cdot \boldsymbol{\sigma}$ with

$$\mathbf{j}_F = \mathbf{j}_\parallel + \cos(2V)\mathbf{j}_\perp - \sin(2V)\mathbf{j}\times\hat{\mathbf{V}}, \quad (44)$$

where $V(k,t) = |\mathbf{V}(k,t)|$, $\hat{\mathbf{V}}(k,t) = \mathbf{V}(k,t)/V(k,t)$, $\mathbf{j}_\parallel(k,t) = \mathbf{j}(k,t)\cdot\hat{\mathbf{V}}(k,t)\hat{\mathbf{V}}(k,t)$ and $\mathbf{j}_\perp(k,t) = \mathbf{j}(k,t) - \mathbf{j}_\parallel(k,t)$ are the components of $\mathbf{j}(k,t)$, respectively, parallel and perpendicular to $\mathbf{V}(k,t)$. The eigenstates of $h_F(k)$ with quasienergies $\pm|\mathbf{d}_F(k)|$ are oriented along $\pm\hat{\mathbf{d}}_F(k,t) = \pm\mathbf{d}_F(k,t)/|\mathbf{d}_F(k,t)|$ on the Bloch sphere.

Thus, after some algebra, we find

$$J(t) = \oint [g_+(k) - g_-(k)]\Big[\mathbf{j}_F(k,t)\cdot\hat{\mathbf{d}}_F(k)\Big]\,dk, \quad (45)$$

where we denote the occupation of $\pm|\mathbf{d}_F(k)|$ quasienergies by $g_\pm(k)$.

We can present explicit expressions for the harmonics for sufficiently small $A_0$, such that $\mathbf{d}_F$ and $\mathbf{V}$ can be approximated well by neglecting $\mathscr{J}_{n>0}(A_0)$. Then, $\mathbf{V} = m_1\mathbf{v}_0\sin(N\omega t)/(N\omega)$ with

$$\mathbf{v}_0 = \Big(1 - \mathscr{J}_0(A_0)\cos k, \mathscr{J}_0(A_0)\sin k, 0\Big), \quad (46)$$

and

$$\mathbf{d}_F = \Big(1 + m_0 + (1 - m_0)\mathscr{J}_0(A_0)\cos k,$$
$$(1 - m_0)\mathscr{J}_0(A_0)\sin k, 0\Big), \quad (47)$$

and Then, $\mathbf{d}_F$, $\mathbf{V}$ and $\mathbf{j}$ are coplanar and $\mathbf{j}\times\hat{\mathbf{V}}\cdot\hat{\mathbf{d}}_F = 0$. So,

$$\mathbf{j}_F\cdot\hat{\mathbf{d}}_F \approx [1-m(t)]\Big\{W_1(k)\sin A(t) + W_2(k)\cos A(t)$$
$$+ \cos(2V)[W_3(k)\sin A(t) + W_4(k)\cos A(t)]\Big\}, \quad (48)$$

where

$$W_1 = +\frac{\mathbf{v}_0\cdot\mathbf{d}_F}{v_0^2 d_F}[\cos k - \mathscr{J}_0(A_0)\cos 2k], \quad (49a)$$

$$W_2 = -\frac{\mathbf{v}_0\cdot\mathbf{d}_F}{v_0^2 d_F}[\sin k - \mathscr{J}_0(A_0)\sin 2k], \quad (49b)$$

$$W_3 = \frac{(1+m_0)\cos k + (1-m_0)\mathscr{J}_0(A_0)}{d_F} - W_1, \quad (49c)$$

$$W_4 = -\frac{(1+m_0)\sin k}{d_F} - W_2, \quad (49d)$$

and

$$v_0^2 = 1 + \mathscr{J}_0^2(A_0) - 2\mathscr{J}_0(A_0)\cos k, \quad (50)$$
$$d_F^2 = (1+m_0)^2 + (1-m_0)^2\mathscr{J}_0^2(A_0)$$
$$+ 2(1-m_0^2)\mathscr{J}_0(A_0)\cos k. \quad (51)$$
$$\mathbf{v}_0\cdot\mathbf{d}_F = 1 + m_0[1 + \mathscr{J}_0^2(A_0)\cos 2k - 2\mathscr{J}_0(A_0)\cos k]$$
$$- \mathscr{J}_0^2(A_0)\cos 2k. \quad (52)$$

Altogether, we find $J^{(p)} = \oint[g_+(k) - g_-(k)]J^{(p)}(k)dk$,

$$J^{(p)}(k) = (1-m_0)\left[\tilde{\mathscr{J}}_p(A_0)\tilde{W}_1 + \sum_{n\in\mathbb{Z}}\mathscr{J}_{2n}(2\tilde{v}_0)\tilde{\mathscr{J}}_{p-2nN}(A_0)\tilde{W}_2\right]$$
$$- \frac{m_1}{2}\sum_{q=\pm N}\left[\tilde{\mathscr{J}}_{p-q}(A_0)\tilde{W}_3 + \sum_{n\in\mathbb{Z}}\mathscr{J}_{2n}(2\tilde{v}_0)\tilde{\mathscr{J}}_{p-2nN-q}(A_0)\tilde{W}_4\right], \quad (53)$$

where we use the shorthands $\tilde{v}_0 = m_1 v_0/(N\omega)$, $\tilde{\mathscr{J}}_n(A_0) = \mathscr{J}_n(A_0)e^{-in\theta}$, and $\tilde{W}_j$ are listed in Table. I. In Fig. 2, we compare the harmonics obtained from the numerical integration of the analytical expression in Eq. (53) to the full numerical calculation reported in the next Section for the ideal Floquet occupations, $g_- = 1, g_+ = 0$, showing excellent agreement for sufficiently large frequencies.

## IV. NUMERICAL RESULTS

In this section, we present our numerical results for the current in the driven SSH model obtained by exact diagonalization of Floquet Hamiltonian and time evolution operators. We have used the Floquet Berry curvature method as given in Eq. 19 to numerically compute the current. We have ensured our results converge both in lattice momentum $k$ and time $t$ by taking a mesh in $k$ with 200 points as well as a mesh in $t$ with 256 points for $\Omega = N\omega$ and 512 points for $\omega = n\Omega$. In order to investigate the effects of occupation of Floquet bands we compare our results for the ideal Floquet and thermal occupation projected to Floquet bands.

### A. Ideal Floquet occupation

First, we consider an ideal Floquet occupation where the Floquet state with lowest quasi-energy band is fully occupied and the higher quasi-energy band is fully empty. The quasienergies are plotted in Fig. 3(a)-(c) for the undriven model with $\delta = 0$, and for the driven model with $\delta/w = 0.17$ and two values of drive frequency $\Omega/\omega = 1$ and $\Omega/\omega = 2$. Below, we provide illustrative results for $\Omega = N\omega$ as well as $\omega = n\Omega$ with $N$ and $n$ positive integers.

We show the Floquet current within a period corresponding to the field frequency in Fig. 4(a)-(c) and the absolute values of its harmonics in Fig. 4(d)-(f) for different values of field amplitude $A_0$ and $\theta = 0$. As expected from our symmetry analysis leading to Eq. (29) with $t_I = T/2, t_R = 0$, we find that for the undriven model, Fig. 4(a) and 4(c), and the driven model with $\Omega/\omega = 2$, Fig. 3(d) and 4(f), the Floquet currents at times $t$ and $t + T/2$ have the same magnitude but opposite sign. Consequently, we also see that only odd harmonics are nonzero in these cases. However, in the driven model with $\Omega/\omega = 1$, Fig. 4(b) and 4(e), these selection rules are lifted and all current harmonics $n \neq 0$ become nonzero. Also, we note the emergence of a multi-step plateau in the HHG spectra that for a stronger optical field and lower field frequency, see Fig. B1 in Appendix B.

Fig. 5 shows the gain in harmonics of the current in the driven system for $\Omega = N\omega$. We call this the "Floquet gain" because it becomes available when the system is periodically driven along with the external field. We

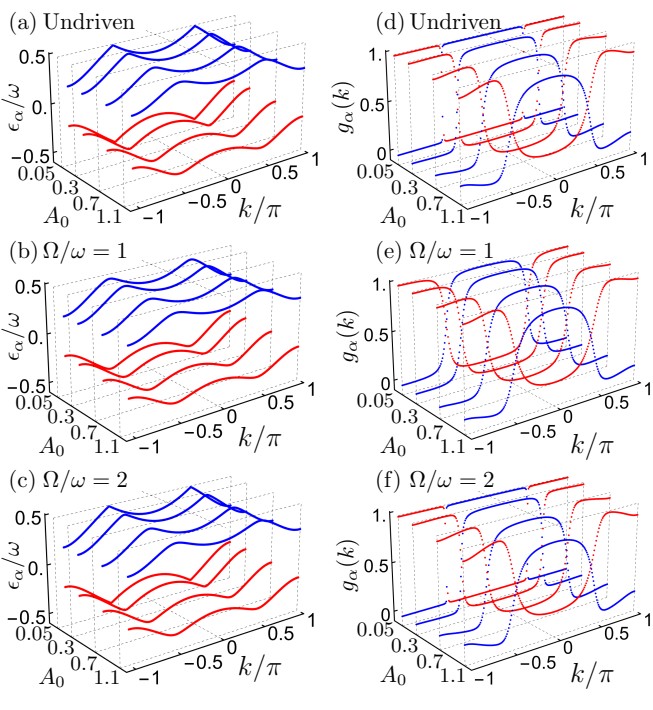

FIG. 3. Quasienergy bands $\epsilon_\alpha(k)$, (a)-(c), of the driven SSH model and their projected thermal occupations $g_\alpha(k)$, (d)-(f). Here, $\omega/w = 3, m_0 = 0.3$ and $\theta = 0$ in all cases, and $m_1 = 0.17$, for the driven cases in (b), (c), (e), and (f). For the undriven cases in (a) and (d) $m_1 = 0$. We note that the spatio-temporal symmetries are the same for the undriven and $\Omega = 2\omega$ cases compared to the $\Omega = \omega$ case. As a result, the first and third rows show very similar features different from the second row, especially around crossings at $k = \pm\pi/2$.

see that as we increase the ratio of drive to field frequencies, we can obtain more and more harmonics. This Floquet gain holds even for low value of the field amplitude as shown in Fig. 5(a). As expected, increasing the field amplitude $A_0$ leads to higher harmonic generation both for undriven and driven cases. Interestingly, the Floquet gain at frequencies $M\Omega \pm |m|\omega$, or equivalently at harmonic order $p = M(\Omega/\omega) \pm |m|$, is prominent even at low field amplitudes. In all panels, and especially in (a), Floquet gain can be observed for $m = 1, 2, \cdots$ up to the nonlinear order with significant value in the undriven model (leftmost column of each panel) and several values of $M = 1, 2, 3$ and 4. Due to the quick suppression of nonlinear contributions with increasing $m$ at low field amplitudes, the Floquet gain can be distinguished easily as streaks of high harmonic generation with slopes equal to $M$ at order $p = M(\Omega/\omega) \pm 1$.

In Fig. 6, we plot the current harmonics as a function of the field amplitude, $A_0$, and the drive amplitude $m_1$. For small amplitudes, the dependence follows a power law with an exponent $\zeta$, which increases as these amplitudes increases. We note that at small field amplitude

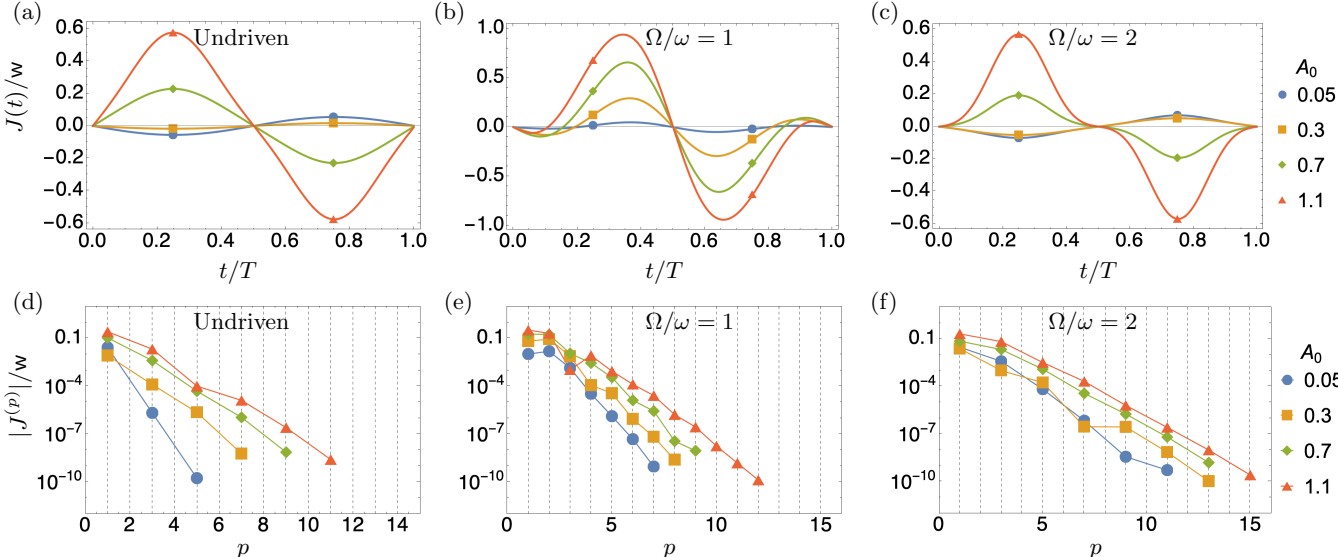

FIG. 4. Optical current and its harmonics in the driven SSH model with ideal Floquet occupation. For different values of field amplitude $A_0$, (a)-(c) show the optical current as a function of time $t$ and (d)-(f) show the absolute value of current harmonics with respect to harmonic order $p$. Here, $\omega/\mathsf{w} = 3$, $m_0 = 0.3$ and $\theta = 0$ for all cases, and $m_1 = 0.17$ for the driven cases.

all HHG components, even those with $p > 1$, show a linear dependence on $A_0$ due to frequency mixing with the drive.

We also investigate the dependence of HHG on the phase difference $\theta - \pi/2$ between the field and the drive. Optical current and its harmonics are calculated numerically and plotted in Fig. 7 for several values of $\theta$. For all $\theta$ when $\Omega = 2\omega$ and, separately, for $\theta = \pi/2$ and $\pi$, we can explicitly observe the symmetries under $I_F$ and $\Theta_F$, respectively, discussed under Eqs. (34). In particular, for $\Omega = 2\omega$ in Fig. 7(b) and (d), the symmetry under $I_F$ forces even harmonics of the current to vanish. Our results show that, while these symmetry properties influence the shape of the current as a function of time, they do not significantly affect the magnitude of its harmonics.

When the drive frequency is a fraction of the field frequency so that $\omega = n\Omega$ for integer $n$, then the optical current for diagonal Floquet occupation produces fractional harmonics of the field frequency. To calculate the fractional harmonics numerically, we consider an ideal Floquet occupation with the lower Floquet band fully occupied. In Fig. 8 we show the absolute value of the harmonics of the current as a function of harmonic order for different field amplitudes. In general, we find harmonics order $p = m + M(\Omega/\omega)$ in agreement with Eq. (12). Of course, we can also view this as the generation of high harmonics of the *drive* frequency. From this perspective, the harmonic generation in the undriven system, Fig. 8(a) provides the amplitudes needed for frequency mixing with the drive.

We note that unlike the previous case, when the drive frequency is the principal Floquet frequency neither the inversion nor the time-reversal symmetry can be restored, since there is no time glide $t \to t + t_0$ within the cycle

for which we can ensure $m(t + t_0) = m(t)$ and $A(t + t_0) = -A(t)$, nor $m(-t + t_0) = m(t)$ and $A(-t + t_0) = -A(t)$. Therefore all the harmonics of the drive that were forbidden in undriven case are now generated.

## B. Projected Floquet thermal occupation

We now compare the optical current harmonics obtained for a thermal density matrix $\rho_0(k) = \frac{1}{Z}e^{-h_0(k)/T_0}$ projected on the Floquet states, where $T_0$ is the temperature (we set $k_B = 1$), $Z = \text{tr}[e^{-h_0(k)/T_0}]$, and the static Hamiltonian $h_0(k) := h(k, t_0)|_{A_0=0, m_1=0}$. That is, we retain only the diagonal elements

$$
\begin{aligned}
g_\alpha(k, t_0) &= \langle \phi_\alpha(k, t_0)|\rho_0(k)|\phi_\alpha(k, t_0)\rangle \\
&= \frac{1}{Z}\sum_\mu e^{-E_\mu^0(k)/T_0}|\langle \phi_\mu^0(k)|\phi_\alpha(k, t_0)\rangle|^2, \quad (54)
\end{aligned}
$$

where $E_\mu^0(k)$ and $|\phi_\mu^0(k)\rangle$ are the energy eigenvalues and eigenstates of $h_0(k)$, respectively. We see the dependence of occupation on $t_0$ remains in Eq. (54) because the Floquet Hamiltonian and, therefore, the Floquet state depend on $t_0$. We would average the current over $t_0$; however, in our numerics we have seen only tiny differences in the occupations of Floquet-Bloch bands for different values of $t_0$. So, we set $t_0 = 0$ for this calculation.

For $T_0 \to 0$, this corresponds to a quenched occupation where the initial state is the ground state of the static Hamiltonian $h_0(k)$ formed by the lowest static energy band $|\phi_{\text{GS}}^0\rangle$, i.e. $g_\alpha(k) = |\langle \phi_{\text{GS}}^0(k)|\phi_\alpha(k)\rangle|^2$. We illustrate the projected Floquet occupation of quasienergy bands in Fig. 3(d)-(f). The Floquet principal frequency in this case is the field frequency $\omega$, which is resonant with

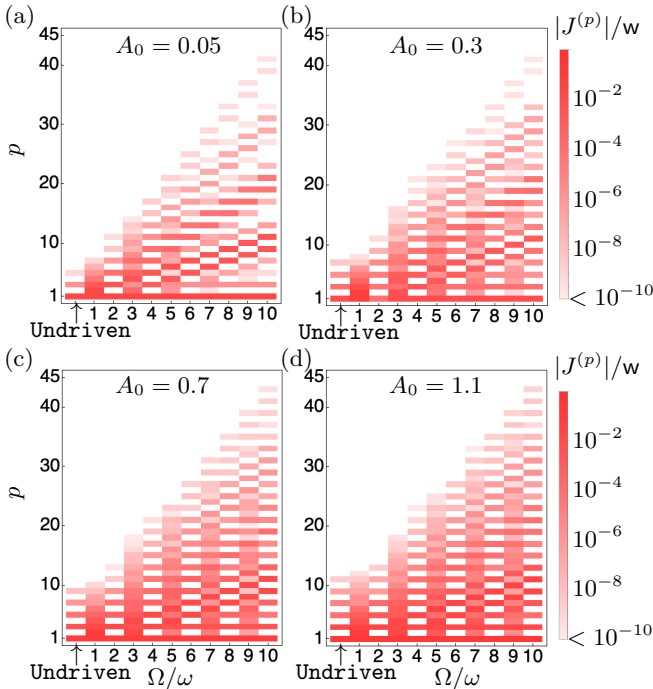

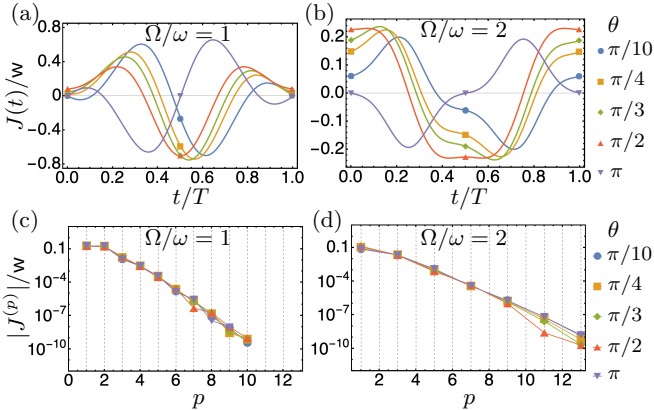

FIG. 7. Dependence of optical current and its harmonics on $\theta$, with $\theta - \pi/2$ as the phase difference between the field and the drive, for ideal Floquet occupation. Here, $\omega/\mathsf{w} = 3$, $m_0 = 0.3$, $m_1 = 0.17$, and $A_0 = 0.71$.

FIG. 5. Floquet gain of the high harmonics of the current in the driven SSH model with ideal Floquet occupation. Here, $\omega/\mathsf{w} = 3$, $m_0 = 0.3$, and $\theta = 0$ for all cases; and $m_1 = 0.17$ for the driven cases. The white background color denotes values too small to show in our precision.

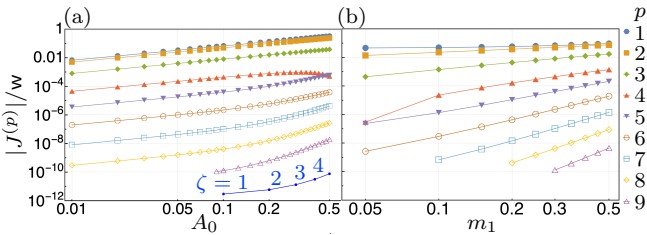

FIG. 6. Current harmonics vs. the field amplitudes (a) and vs. driven hopping term (b) in the driven SSH model with ideal Floquet occupation. Here, $\omega/\mathsf{w} = 3$, $m_0 = 0.5$, $\Omega = \omega$, and $\theta = 0$. We have taken $m_1 = 0.3$ in (a) and $A_0 = 0.1$ in (b). At the bottom of panel (a), we show lines corresponding to power-laws with exponent $\zeta$ for easy comparison.

forbidden harmonics, such as $J^{(0)}$, even for the undriven system. Interestingly, we find significant Floquet gain for $\Omega/\omega = 2$ even at low field amplitudes.

## V. CONCLUSIONS AND OUTLOOK

We have developed a theoretical framework using Floquet theory to calculate the optical current in a periodically driven system. The Floquet optical current mixes the field and drive frequencies, $\omega$ and $\Omega$, respectively, generating harmonics $m\omega + M\Omega$. The frequency mixing broadens the spectrum of harmonic orders and enhances the values of harmonics even for a low field amplitude.

This formulation naturally contains the Floquet theoretic approach to understanding high harmonic generation in initially undriven systems and the Floquet linear response theory [63] as limiting cases when the drive amplitude and the field amplitude, respectively, approach zero. In the latter case this work provides a nonlinear Floquet theoretic framework for optical current for any time periodic field.

The optical current can be recast in terms of the occupation of Floquet-Bloch bands and their non-adiabatic Berry connection and curvature. This formulation naturally brings out a relation between topology and optical current for sufficiently low drive and field frequencies and exposes and clarifies subtleties of gauge fixing that prevent a direct relation between current harmonics and Floquet topological invariants for higher frequencies.

In the presence of spatio-temporal symmetries like inversion and time reversal with time glide, we obtained the properties of optical current and selection rules for the harmonics. As an application of our formulation, we studied the optical current and in the driven SSH model and found analytical expressions for its harmonics at sufficiently high frequency and weak field amplitude. In our example, in the absence of the optical field, the driven

the energy bands $E_\mu^0$. Therefore, there is a qausienergy band inversion and quasienergy gap induced by the field and the drive. Note also that the projected Floquet occupations are not symmetric in $k$ and that the asymmetry increases with with the field amplitude $A_0$.

In Fig. 9, we plot the Floquet current within a drive cycle and show its harmonics . Even though the Hamiltonian is symmetric under $I_F$ and $\Theta_F$, the asymmetry of the projected Floquet occupation under $k \to -k$ now breaks the corresponding selection rules of the optical current. Therefore, we obtain many of the previously

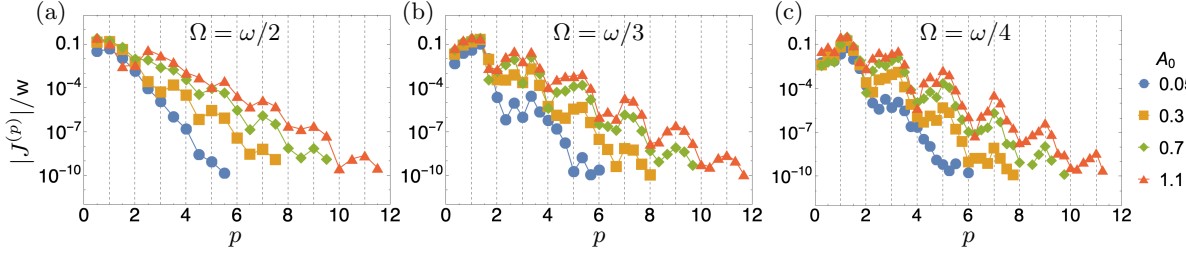

FIG. 8. Fractional harmonics of the current in the driven SSH model with ideal Floquet occupation. Here, $\omega/\mathsf{w} = 3, m_0 = 0.3, m_1 = 0.17$, and $\theta = 0$.

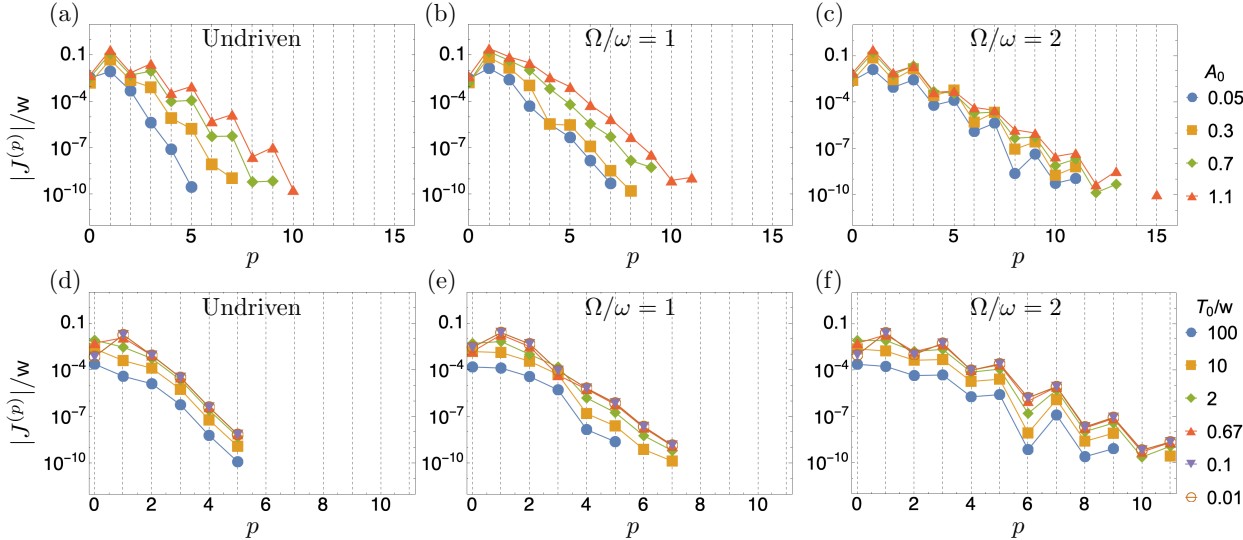

FIG. 9. Current harmonics for the projected Floquet thermal occupation for different field amplitudes $A_0$ (a)-(c), and for different temperatures $T_0$ (d)-(f). Here, $\omega/\mathsf{w} = 3, m_0 = 0.3$ and $\theta = 0$ for all cases, and $m_1 = 0.17$ for the driven cases. We have taken $T_0/\mathsf{w} = 0.01$ in (a)-(c), and $A_0 = 0.1$ in (d)-(f).

hopping term by itself produces no current and, therefore, no HHG. However, the driven hopping term helps in broadening the harmonic spectra of the optical drive. Hence the two drives in our example are different from the standard two-color drives where each drive itself produces current and harmonics. Symmetry analysis and numerical calculations show that by altering the temporal part of the spatio-temporal symmetry, we can obtain harmonics that are forbidden in undriven cases. The broader implication of this phenomenon is that without changing the spatial structure or symmetry of a material, we can procure forbidden harmonics by driving the system.

Importantly, frequency mixing between the drive and the field opens a way to enhance the HHG spectrum. This Floquet enhancement can be accessed naturally in pump-probe setups, where the pump and drive frequencies are commensurate. For non-optical drives, a more realistic setup is when the drive frequency is a small commensurate fraction of the field frequency, which always breaks the temporal symmetry and leads to a broad-spectrum HHG. Additionally, our calculations pertaining to the driven SSH model may be tested in synthetic quantum simulators. [73]

Our results demonstrates that we can broaden and enhance the spectrum of harmonic order by driving the system. This work encourages further theoretical and experimental investigations of harmonic generation in bulk driven quantum systems.

## ACKNOWLEDGMENTS

This work has been supported in part by the NSF CAREER award DMR-1350663, the U.S. Department of Energy, Office of Science, Basic Energy Sciences, under Award No. DE-SC0020343, and the Vice Provost for Research at Indiana University, Bloomington through the Faculty Research Support Program.

## Appendix A: Symmetries of the Floquet Hamiltonian

In this appendix, we derive the symmetries of the current from fully occupied Floquet bands, Eqs. (24) and (26), under the unitary mirror and antiunitary time-reversal symmetries, Eqs. (22) and (23), respectively.

The unitary symmetry $I_F$ transforms the Floquet unitary $U_F(k,t) = \mathrm{Texp}[-i\int_t^{t+T} h(k,s)ds] = e^{-iTh_F(k,t)}$ as

$$I_F U_F(k,t) I_F^\dagger = \mathrm{Texp}\left[-i\int h(-k, s+t_I)ds\right] = U_F(-k, t+t_I). \tag{A1}$$

Thus, $I_F^\dagger h_F(-k, t+t_I) I_F = h_F(k,t)$ and the projector $P_{\mathrm{occ}}(k,t) = \sum_{\alpha\in\mathrm{occ}} |\phi_\alpha(k,t)\rangle\langle\phi_\alpha(k,t)|$ onto the occupied Floquet bands is mapped as

$$I_F P_{\mathrm{occ}}(k,t) I_F^\dagger = P_{\mathrm{occ}}(-k, t+t_I). \tag{A2}$$

The occupied Floquet states are mapped explicitly as $I_F |\phi_\alpha(k,t)\rangle = \sum_{\beta\in\mathrm{occ}} \mathfrak{I}_{\beta\alpha}(k,t) |\phi_\alpha(-k, t+t_I)\rangle$, with a unitary matrix $\mathfrak{I}(k,t)$: for $\lambda, \beta \in \mathrm{occ}$,

$$
\begin{aligned}
[\mathfrak{I}(k,t)\mathfrak{I}^\dagger(k,t)]_{\lambda\beta} &= \sum_{\alpha\in\mathrm{occ}} \mathfrak{I}_{\lambda\alpha}(k,t)\mathfrak{I}_{\beta\alpha}^*(k,t) \\
&= \sum_{\alpha\in\mathrm{occ}} \langle\phi_\lambda(-k, t+t_I)|I_F|\phi_\alpha(k,t)\rangle \langle\phi_\beta(-k, t+t_I)|I_F|\phi_\alpha(k,t)\rangle^* \\
&= \sum_{\alpha\in\mathrm{occ}} \langle\phi_\lambda(-k, t+t_I)|I_F|\phi_\alpha(k,t)\rangle \langle\phi_\alpha(k,t)|I_F^\dagger|\phi_\beta(-k, t+t_I)\rangle \\
&= \langle\phi_\lambda(-k, t+t_I)|I_F P_{\mathrm{occ}}(k,t) I_F^\dagger|\phi_\beta(-k, t+t_I)\rangle \\
&= \langle\phi_\lambda(-k, t+t_I)|P_{\mathrm{occ}}(-k, t+t_I)|\phi_\beta(-k, t+t_I)\rangle \\
&= \delta_{\lambda\beta}. 
\end{aligned}
\tag{A3}
$$

Then, the current has the symmetry

$$
\begin{aligned}
J(t) &= \sum_{\alpha\in\mathrm{occ}} \oint \langle\phi_\alpha(k,t)|\partial_k h(k,t)|\phi_\alpha(k,t)\rangle\, dk = \oint \mathrm{tr}[P_{\mathrm{occ}}(k,t)\partial_k h(k,t)]dk \\
&= \oint \mathrm{tr}[P_{\mathrm{occ}}(k,t) I_F^\dagger \partial_k h(-k, t+t_I) I_F]dk = \oint \mathrm{tr}[I_F P_{\mathrm{occ}}(k,t) I_F^\dagger \partial_k h(-k, t+t_I)]dk \\
&= \oint \mathrm{tr}[P_{\mathrm{occ}}(-k, t+t_I)\partial_k h(-k, t+t_I)]dk = -\oint \mathrm{tr}[P_{\mathrm{occ}}(k, t+t_I)\partial_k h(k, t+t_I)]dk \\
&= -J(t+t_I). 
\end{aligned}
\tag{A4}
$$

We note that the current vanishes when $t_I = 0$. This happens in our driven SSH model when the optical filed is absent, $A(t) = 0$. Therefore, driving the hopping term alone cannot produce a current in our example under the ideal Floquet occupation.

The antiunitary symmetry $\Theta_F$ transforms the Floquet unitary as

$$
\begin{aligned}
\Theta_F U_F(k,t) \Theta_F^\dagger &= \prod_{s:t\to t+T} \Theta_F e^{-ih(k,s)ds} \Theta_F^\dagger \\
&= \prod_{s:t\to t+T} e^{+ih(-k, -s+t_R)ds} \\
&= \left(\prod_{s:-t-T+t_R\to -t+t_R} e^{-ih(-k,s)ds}\right)^\dagger \\
&= U_F^\dagger(-k, -t+t_R). 
\end{aligned}
\tag{A5}
$$

Here, $\prod_{s:t_1\to t_2}$ denotes the time-ordered product over $t_1 < t_2$. Thus, $\Theta_F^\dagger h_F(-k, -t+t_R)\Theta_F = h_F(k,t)$ and

$$\Theta_F P_{\mathrm{occ}}(k,t) \Theta_F^\dagger = P_{\mathrm{occ}}(-k, -t+t_R). \tag{A6}$$

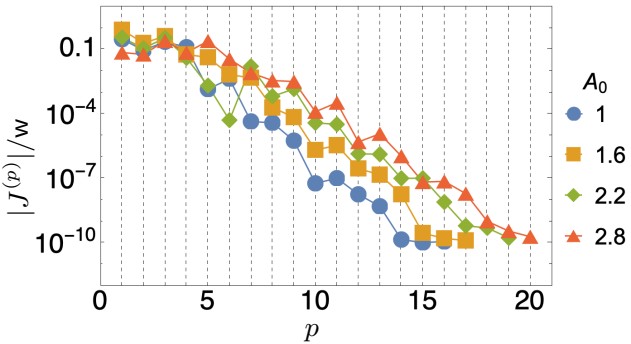

FIG. B1. Plateau in the harmonics at low frequency and strong filed amplitude under ideal Floquet occupation. Here, $\omega/\mathsf{w} = 1.2$, $m_0 = 0.3$, $\Omega = \omega$, $\theta = 0$, and $m_1 = 0.17$.

Explicitly, the occupied Floquet states are mapped as $\Theta_F |\phi_\alpha(k,t)\rangle = \sum_{\beta \in \text{occ}} \mathfrak{T}_{\beta\alpha}(k,t) |\phi_\alpha(-k, -t+t_R)\rangle$, with an orthogonal matrix $\mathfrak{T}(k,t)$: for $\lambda, \beta \in$ occ,

$$
\begin{aligned}
[\mathfrak{T}(k,t)\mathfrak{T}^{\mathsf{T}}(k,t)]_{\lambda\beta} &= \sum_{\alpha \in \text{occ}} \mathfrak{T}_{\lambda\alpha}(k,t)\mathfrak{T}_{\beta\alpha}(k,t) \\
&= \sum_{\alpha \in \text{occ}} \langle\phi_\lambda(-k,-t+t_R)|\Theta_F|\phi_\alpha(k,t)\rangle \langle\phi_\beta(-k,-t+t_R)|\Theta_F|\phi_\alpha(k,t)\rangle \\
&= \sum_{\alpha \in \text{occ}} \langle\phi_\lambda(-k,-t+t_R)|\Theta_F|\phi_\alpha(k,t)\rangle \langle\phi_\alpha(k,t)|\Theta_F^\dagger|\phi_\beta(-k,-t+t_I)\rangle \\
&= \langle\phi_\lambda(-k,-t+t_R)|\Theta_F P_{\text{occ}}(k,t)\Theta_F^\dagger|\phi_\beta(-k,-t+t_R)\rangle \\
&= \langle\phi_\lambda(-k,-t+t_R)|P_{\text{occ}}(-k,-t+t_R)|\phi_\beta(-k,-t+t_R)\rangle \\
&= \delta_{\lambda\beta}, \tag{A7}
\end{aligned}
$$

where we have used the identity $\langle\psi|\Theta_F|\chi\rangle = \langle\chi|\Theta_F^\dagger|\psi\rangle$ for antiunitary $\Theta_F$. Similarly, the current has the symmetry

$$
\begin{aligned}
J(t) &= \oint \text{tr}[P_{\text{occ}}(k,t)\partial_k h(k,t)]dk \\
&= \oint \text{tr}[P_{\text{occ}}(k,t)\Theta_F^\dagger \partial_k h(-k,-t+t_R)\Theta_F]dk = \oint \text{tr}[\Theta_F P_{\text{occ}}(k,t)\Theta_F^\dagger \partial_k h(-k,-t+t_R)]dk \\
&= \oint \text{tr}[P_{\text{occ}}(-k,-t+t_R)\partial_k h(-k,-t+t_R)]dk = -\oint \text{tr}[P_{\text{occ}}(k,-t+t_R)\partial_k h(k,-t+t_R)]dk \\
&= -J(-t+t_R), \tag{A8}
\end{aligned}
$$

as stated in he main text.

## Appendix B: HHG Plateaux at Lower Frequency and Stronger Field Amplitude

In Fig. B1, we plot the HHG harmonics for smaller field frequency and larger values of the field amplitude. We can now see a prominent multi-step plateau. Especially for $A_0 = 2.8$, the first plateau can be seen from harmonic 1 to 5, and the next plateaus from 7 to 9, and from 15 to 17 and so on. This multi-step plateau reflects the growing effects in the non-perturbative regime.

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

arXiv:2101.02871 [physics.atom-ph].