# Peer review of "Enhancement of High Harmonic Generation in Bulk Floquet Systems"

_SciPost Physics_

## Round 1 · Referee Report · Anonymous · 2023-7-24

Report

In the new revised text, the authors have addressed some of my concerns, and answered my questions. But in my opinion, they have not given enough effort to addressing the issues raised. I think that we are basically still in a disagreement regarding the novelty of some of the claims in the text. Specifically, the idea of selection rules arising from spatio-temporal dynamical symmetries, and, the degree of innovation and significance for those in implementations of HHG from Floquet-driven systems. Indeed, from my perspective, if I read the abstract and introduction of the revised text, I still think it miss-represents prior work and over claims the current results. The background regarding other works on selection rules from systems driven by an arbitrary number of pumps is only introduced in the main text rather than in the introduction, and still not put in proper context. Moreover, the authors write that no work has considered time-reversal symmetries/time-translation/spatial-translation in the context of selection rules in their responses. This is not correct. Time-reversal has been considered first in Nat. Comm. 10, 405 (2019) (still not cited), and in some follow up works such as the current ref. 69, Sci. Adv. 9, eade0953 (2023), and even in PRL 127, 126601 (2021) in the context of solids driven by two pumps (which obviously then includes also spatial translation in the context the authors mean, although that one is irrelevant for the symmetry-based selection rule). There are other works in solids, and in macroscopic systems. The idea that with two drives one can break the system’s time-translation symmetry and see ‘forbidden’ harmonics is well-established in many other systems and conditions, and in solids, as I already stated in my previous report. It is in fact the basis for symmetry-breaking spectroscopies.
Another thing that has become apparent to me in this round is that the authors artificially separate the two drives, with one purely driving the electronic system, and one modulating the hopping amplitudes, inducing the Floquet phase. On the one hand, this separation distinguishes their work from previous literature, as usually two laser pumps are considered rather than directly tuning the hopping amplitudes. On the other hand, this reduces the degree of potential impact of this work, because in experimental implementations the only realistic way to actually tune these parameters is to add another electromagnetic drive. Of course, there are some other options, e.g. adding a time-periodic magnetic field, or periodically tuning some other parameter of the system, but to get to those frequencies employed by the authors typically one has to use some form of light. This is most apparent in my question to the authors about the case where both frequencies overlap and are equal to w. In that case it is clear that there is no way to tune the hopping amplitudes without driving with another optical field at w, making the two fields in fact identical and overlapping in time and space, and in any other formalism one could just write them as a single optical field at w. In that case it’s also clear that the way the authors calculate any HHG ‘enhancement’ factors is not a ‘fair’ estimate, because there are two drives at w, and one has to compare the HHG yield emitted from the Floquet system to that emitted from the non-Floquet system driven at an amplitude E0+E_floquet. This issue of course persists beyond the case where both drives have equal frequencies, and it’s general to the text. It’s also quite crucial in my mind to elucidate, as even in the title this yield enhancement is claimed.
From a methodological point of view, the statement that the Floquet drive does not generate a current, or emit harmonics, does not make sense to me. Again, the system has to be coupled to some external perturbation to generate that Floquet phase (likely an electromagnetic field), and that will probably generate a current. Even THz drives are known to generate harmonics (as cited by the authors).
Looking at it as a whole, from the Scipost Physics website I read the acceptance criteria. I think that all ‘general acceptance criteria’ can be upheld in a revised submission, once the work is put in proper context as I suggested. However, I don’t think that this paper upholds any one of the ‘Expectations’, and could not be revised to do so. Thus, I cannot recommend publication in Scipost Physics. I would support a revised version to be published in the less stringent SciPost Physics Core, pending revisions by the authors that put things into the proper context and address the issue of defining the yield enhancement.

---

## Round 1 · Referee Report · Anonymous · 2023-7-31

Report

The authors changed very little in the revised version. Their replies to my few points of criticism are far from satisfactory. Claiming HHG in Floquet systems without showing non-perturbative HHG spectra and mentioning a possible realisation is not acceptable. Their vague idea about phonons is only mentioned in the reply but not in the revised manuscript.

Also the way the authors cite the suggested references 66-69 is not acceptable, since 66-69 also discuss spatio-temporal symmetries. The authors suggest novelty where there is none.

I also still do not get the message of Fig. 3. The differences are too minuscule, and it is the job of the authors to present them appropriately and to explain why these tiny differences are relevant.

I do not recommend publication of that manuscript.

---

## Round 1 · Author Response

Dear Editor,

We have considered the reports carefully and provide a full response to all the points raised by the Referees. We have revised the manuscript accordingly. We believe this has improved our presentation and hope you find it suitable for publication.

Referee 1: We thank the referee for the careful review and nice summary of the manuscript. This gives us confidence that our presentation is clear and we are pleased to receive the referee’s recommendation for publication. We provide responses to Referee’s suggestions and comments below.

  1. How could HHG in a Floquet system be observed? The kind of driving the authors consider is possible to implement on various platforms that allow to simulate tight-binding hamiltonians, e.g., cold atoms in optical lattices, optically, with written wave guides, or even classically with electric circuits. However, in all such systems no high harmonics of incident light are emitted. HHG occurs in "real" physical systems that sustain the strong incident light. One could drive such a system with a second laser but that would be completely different from the driven hopping amplitudes considered in the manuscript. It would rather be HHG in two-color fields, which has been studied extensively already. The authors might discuss which experimental realization they envision

Reply: We agree with the referee that having concrete material platforms would be desirable. There are many considerations that need to be made in choosing a more realistic setting, especially for solid-state systems, such as available drives, possible ranges of drive amplitude and frequency, heating, etc. This is beyond the scope of this work, but we may speculate about possible realization of the type of drive we consider (of the hopping amplitudes) in phonon driven Floquet matter (ref: Nano Lett. 2018, 1535-1542) when irradiated by light or mechanical drives (e.g. time periodic strain or cycle of stretch and relaxation) that may be developed for a 2D system; however, any such proposal needs to be studied in detail in future work.

  1. Concerning selection rules for HHG using Floquet theory, the authors may consult, https://doi.org/10.1103/PhysRevLett.80.3743, https://doi.org/10.1088/0953-4075/34/24/305, https://doi.org/10.1103/PhysRevA.91.053811, for instance.

Reply: Thank you for bringing these papers to our attention. We have cited those in our revised version.

  1. Figure 3 shows three times the same set of two panels. I wonder how such a mistake remains unnoticed by three authors.

Reply: Three figures are actually not the same: there are differences among them, which seem to have been too small to notice without pointing out. We considered modifying the figures, removing panels, etc. but eventually decided the information contained in the original figure was important to communicate and have instead added a note in the caption to clarify and compare the information in different panels.

  1. In general, the manuscript is well written but there are still some typos. I leave it as an exercise for the authors to figure them out.

Reply: Thank you! We have fixed the typos to the best of our findings.

Referee 2:

We thank the referee for reviewing the manuscript in detail. We think that most of the questions raised by the referee stem from an underlying question regarding the nature the drives. We note here that in our study the two drives are not both optical in nature. For example, in the driven SSH model, the Floquet drive is in the hopping terms, while the HHG probe is optical. Our formalism works for any two periodic drives. However, without an optical drive (that couples to the phase of the hopping amplitude) there will be no current in the system.

In the following, we have answered the questions as well as listed the changes made in the revised manuscripts. These changes further improve the clarity and readability of the paper. We hope that our responses and changes satisfy the referee to recommend this manuscript for publication. Weaknesses

  1. I could not understand mainly what is the fundamental difference between utilizing this approach (one color for Floquet physics + one for an HHG drive), or just having a standard two-color driving fields. Both induce a time-periodic Hamiltonian, and both can be analyzed with Floquet theory. In fact, it was difficult for me to follow if the current approach by the authors included also the optical drive in the Floquet theorem, or only the 2nd drive at $\Omega$ for the hopping terms.

In that respect, it's not clear if this is just a mathematical choice for analyzing the system for convenience, or if there's some added intuition, because one could just as easily analyze HHG results with multi-colored drives for various solids, as is in fact very commonly done with many other theoretical frameworks. I suppose this version allows extracting some high-frequency limit, but I'm not sure that's the most relevant thing (see point 2).

Reply: The fundamental difference between the standard two-color driving approach and ours is that our formalism accommodates all sorts of periodic drives, be it optical or mechanical or something else. For example, we have included both optical and hopping drives in our treatment of the driven SSH model. It is important to note that not all types of the periodic drives can produce current or harmonics. In our example, the time-periodic hopping term alone does not produce any current in the system. Rather, it helps in the enhancement and broadening of the harmonic spectra of the optical drive. This is in contrast to the multi-color drives where each drive alone can produce current or harmonics. So, an analysis based on the multi-color drive can not describe our example directly. However, our Floquet formalism can, in principle, be used for HHG due to multi-color drives as well.

  1. A big portion of the calculations and analysis is performed for high frequencies. This also means the response is not necessarily non-perturbative. Most plots in the paper show quite strong exponential decay of harmonics with harmonic order (e.g. figs. 2,4,9, etc.), and no plateau. Thus, I would make a separation, because the standard notation for HHG is for truly non-perturbative optical responses. I would either focus the work on perturbative responses, or present spectra that show plateaus. Also in this regard, a lot of the introduction references are in fact only dealing with SHG and perturbative responses.

Reply: Our general formalism and numerical calculations are valid for all frequency and amplitude strengths. Additionally, even high frequency approximation is only perturbative in inverse frequency, and non-perturbative in terms of the strength of the field amplitude. However, for numerical convenience and for illustration of the effect we have chosen the relatively weak field amplitude.

To clarify the point further, we have added a figure in the appendix to show broadening and development of a plateau at lower frequency and stronger field amplitude.

  1. The analysis and conclusions on the spatio-temporal symmetries and selection rules overlaps with many years of efforts already well-established in the strong-field physics community. It is hard to summarize this effort, but one can look at some main papers, including Phys. Rev. Lett. 80, 3743 (1998), J. Phys. B: At. 34, 5017 (2001), and more recent generalizations that also include time-reversal symmetries and glide symmetries explored by the authors (Nat. Comm. 10, 405 (2019) , Nat. Comm. 13, 1312 (2022)). This approach has already been employed to many HHG setups from atoms and solids, and even recently was used to analyze HHG from topological insulators (Phys. Rev. A 103, 023101 (2021), Nano Lett. 21, 8970–8978 (2021), again, its hard to summarize because its already very well established). It is well known that by employing an additional commensurate field one can control (break/impose) symmetries in the system. This is even the case in EFISH, which is a subcase of Floquet theory, but also its analogue in HHG (Nat. Phot. 12, 465–468 (2018)). Here the authors have to acknowledge previous works, including the most general works and remove claims of novelty. Essentially, I would say the authors re-derive a lot of these results with new terminology, but there is no new physics presented since the selection rules arise from Floquet group theory.

Reply: We acknowledge that there are several works in the literature related to selection rules for high harmonics based on symmetries. We have cited them and we thank the referee for bringing additional papers to our attention.

However, we must also point out that our discussion of selection rules is still new and differs from those found in previous literature in some important ways. For example, none of the referenced papers have considered both discrete spatial and time translational symmetries together or some special symmetries like time-reversal. Even the most general studies on the selection rules based on the Floquet group (Nat. Comm. 10, 405 (2019) , Nat. Comm. 13, 1312 (2022)), have only considered the time translational symmetries but not the spatial and translational symmetry together.

Our formalism starts with the Floquet-Bloch Hamiltonian and therefore it naturally contains both discrete spatial and time translational symmetries. This is important since, as we have already emphasized, a general periodic drive may not produce any current by itself, yet it can enhance HHG and broaden the harmonic spectra. A manifestation of this in our work is that without changing the optical field itself (which is the drive that produces current and harmonics), we can break the temporal part of the symmetries by changing the periodic hopping parameter.

  1. The authors claim there is an enhancement of the HHG emission in the driven systems. I suppose this is quite expected. But, I think the enhancement has not been evaluated fairly here - one should compare the HHG yields not between the driven/undriven systems, but between the driven system, and an undriven system where the HHG generating field has a total power that corresponds to the total driving power in the driven system. In other words, the power of the 2nd field at $\Omega$ should be put back into the fundamental field for correct comparison, otherwise the two cases are not on equal footing.

Reply: We welcome the suggestion of the referee for comparison of the harmonics in terms of the yield. However, we do not think simply adding the powers of the Floquet and optical drives provides the appropriate measure because, in our case, the Floquet drive (the periodically driven hopping term) does not produce any current or harmonics when acting alone and, therefore, its effect on HHG can only be seen in the presence of the optical drive in the system. Thus, the total power is not a good measure of comparison. Instead, our characterization in terms of the driven/undriven shows the enhancement and broadening of the harmonic spectra in the presence/absence of the time periodic hopping term, which is mainly responsible for dressing the bands.

  1. Another weakness is that the entire study is done for a 1D model system, such that one cannot make general conclusions for more realistic systems.

Reply: Since the coefficient of harmonics depends on the occupation and details of the Hamiltonian, we have considered the 1D model for our numerical demonstration of enhancement and broadening of harmonic spectra. Our theoretical formalism can be easily extended and used to study higher dimensional models by taking the quasi-momentum as a higher dimensional vector. This is already pointed out in Sec. IIC.

Report

  1. The HHG process is itself a Floquet process that can 'drive' a system out of equilibrium. For instance, this was predicted to cause a topological phase transition (Nat. Phot. 14, 728-732 (2020)), a dynamical band dressing (Nat. Phot. 16, 428-432 (2022)), and some other related phenomena, and there's some debate on what picture should be used for interpreting the dynamics (Phys. Rev. Research 4, 033101 (2022)), i.e. the dressed band picture or the field-free bands. I think the authors should comment on this, because it raises the question of which field should be considered the probe and which the pump, and how would one choose which Floquet basis to use if there are two drives, etc.

Reply: For weak optical drive, one could perform a perturbative expansion for nonlinear transport coefficients. This follows from our formalism directly, for example, by expanding the Bessel functions in Sec. IIIB. From this perspective, one might consider the optical drive as the probe. However, our work will be most useful when the two drives (regardless of their nature) modify the bands of the bulk solid sufficiently to create dressed quasi-energy bands. Because of this, both drives should be considered in the Floquet formalism. In a pump-probe experiment, we may consider an additional optical drive for the probe. Finding the exact field strength up to which adiabatic bands could work (as done in Phys. Rev. Research 4, 033101 (2022) ) is beyond the scope of this paper.

  1. The authors should also address in the introduction Phys. Rev. B 99, 195428 (2019), Phys. Rev. A 99, 053402 (2019), Phys. Rev. A 102, 053112 (2020), which explore similar physics but in non-dressed systems.

Reply:​ We thank the referee for bringing these papers to our attention.

  1. The notation for curly omega and omega is confusing (e.g. in fig. 2). I would make an effort to change notations and clarify some of the variables.

Reply: In fig 2, one term is $w$ (overall hopping parameter, see eq.(33)) and the other is frequency $\omega$. We have replaced the hopping unit $w$ by sans-serif $\mathsf{w}$ to bring more distinction with $\omega$, the probe frequency.

  1. I wonder how can one control the occupations of the Floquet states in a realistic system - would this not be determined by the dephasing properties of the solid?

Reply: In a realistic system, the occupations of Floquet states will be determined by how the system is coupled with the environment. A possible mechanism could be based on decoherence, where some degrees of the freedom of the system couple (entangle) with the environment leading to a particular type of occupation. However, the non-equilibrium occupations and the question of their control are still open questions and beyond the scope of this work.

  1. is $t_0$ averaged over one drive cycle? or more? because it can often take a few driving cycles for the system to enter into a so-called steady-state.

Reply: In our numerical calculations, we have chosen the diagonal Floquet occupation which can be thought as averaged over $t_0$ when the occupation g is independent of $t_0$. In practice, one may wish to do this averaging differently and also include the pulse shape of the drive, etc. Again, these considerations have more to do with specific realizations and are beyond the scope of this work.

  1. When both drives are of the same frequency, labeled $\Omega=\omega$, what then is the meaning of the 2nd drive? Would it not be equivalent to a case with just a single drive?

Reply: In this case, the difference between the two drives comes down to their nature, i.e. the way they couple to the system’s degrees of freedom. As we clarified in previous responses, the optical probe is the one responsible for generating current, while the Floquet drive could be of an entirely different nature.

---

## Round 1 · List of Changes

List of changes in the revised manuscript:
Some changes are made in the text of the manuscript (given in blue color)
We have cited the papers suggested by the referees.
We have added a note in the caption of the figure 3.
We have added a figure in the appendix to show a developing plateau at strong field amplitude.
We have replaced the hopping unit $w$ by sans-serif $\mathsf{w}$ to bring more distinction with $\omega$, the probe frequency.
Miscellaneous typos corrected.

---

## Editorial Decision

awaiting_resubmission